# Agomelatine: A Potential Multitarget Compound for Neurodevelopmental Disorders

**DOI:** 10.3390/brainsci13050734

**Published:** 2023-04-27

**Authors:** Rosa Savino, Anna Nunzia Polito, Gabriella Marsala, Antonio Ventriglio, Melanie Di Salvatore, Maria Ida De Stefano, Anna Valenzano, Luigi Marinaccio, Antonello Bellomo, Giuseppe Cibelli, Marcellino Monda, Vincenzo Monda, Antonietta Messina, Rita Polito, Marco Carotenuto, Giovanni Messina

**Affiliations:** 1Department of Woman and Child, Neuropsychiatry for Child and Adolescent Unit, General Hospital “Riuniti” of Foggia, 71122 Foggia, Italy; 2Drug’s Department ASP Catania, 95100 Catania, Italy; 3Department of Clinical and Experimental Medicine, University of Foggia, 71122 Foggia, Italy; 4Department of Experimental Medicine, Section of Human Physiology and Unit of Dietetics and Sports Medicine, Università degli Studi della Campania “Luigi Vanvitelli”, 80131 Naples, Italy; 5Department of Movement Sciences and Wellbeing, University of Naples “Parthenope”, 80133 Naples, Italy; 6Department of Mental and Physical Health and Preventive Medicine, Università degli Studi della Campania “Luigi Vanvitelli”, 80131 Naples, Italy

**Keywords:** AGM, melatonin agonist and selective serotonin antagonist (MASS), autism spectrum disorder (ASD), attention deficit hyperactivity disorder (ADHD), neurodevelopmental disorders

## Abstract

Agomelatine (AGM) is one of the latest atypical antidepressants, prescribed exclusively for the treatment of depression in adults. AGM belongs to the pharmaceutical class of melatonin agonist and selective serotonin antagonist (“MASS”), as it acts both as a selective agonist of melatonin receptors MT1 and MT2, and as a selective antagonist of 5-HT2C/5-HT2B receptors. AGM is involved in the resynchronization of interrupted circadian rhythms, with beneficial effects on sleep patterns, while antagonism on serotonin receptors increases the availability of norepinephrine and dopamine in the prefrontal cortex, with an antidepressant and nootropic effect. The use of AGM in the pediatric population is limited by the scarcity of data. In addition, few studies and case reports have been published on the use of AGM in patients with attention deficit and hyperactivity disorder (ADHD) and autism spectrum disorder (ASD). Considering this evidence, the purpose of this review is to report the potential role of AGM in neurological developmental disorders. AGM would increase the expression of the cytoskeleton-associated protein (ARC) in the prefrontal cortex, with optimization of learning, long-term memory consolidation, and improved survival of neurons. Another important feature of AGM is the ability to modulate glutamatergic neurotransmission in regions associated with mood and cognition. With its synergistic activity a melatoninergic agonist and an antagonist of 5-HT2C, AGM acts as an antidepressant, psychostimulant, and promoter of neuronal plasticity, regulating cognitive symptoms, resynchronizing circadian rhythms in patients with autism, ADHD, anxiety, and depression. Given its good tolerability and good compliance, it could potentially be administered to adolescents and children.

## 1. Introduction

Agomelatine (AGM) is a melatonin analog with antidepressant properties prescribed for the treatment of depression in adults. It was approved by the European Medicines Agency (EMA) in 2009 and the Therapeutic Goods Administration in Australia in 2010 [1,2]. This molecule is also effective in the treatment of generalized anxiety disorder (GAD) [2,3,4], as well as in bipolar depression, alcohol abuse, and migraines [2,5,6].

Due to the paucity of clinical trials in the pediatric population, the safety and efficacy of AGM in children and adolescents have not yet been established.

### AGM Pharmacology

AGM belongs to the pharmaceutical class of melatonin agonist and selective serotonin antagonist (MASS) since it acts synergistically as a selective agonist of the melatoninergic receptors MT_1_ and MT_2_, and a selective antagonist of 5-HT2C/5-HT2B receptors [7]. As an agonist of MT_1_/MT_2_, AGM has positive effects on the sleep–wake cycle, while as an antagonist of postsynaptic 5-HT2C receptors, it increases the availability of norepinephrine and dopamine in the prefrontal cortex with an antidepressant and nootropic effect [8]. In addition, antagonism of 5-HT2C may modulate the GABAergic pathway by the activation of GABA neurons in the amygdala, bed nucleus of stria terminalis, and hippocampus, with an anxiolytic activity [1,7].

After oral administration, AGM is rapidly absorbed (>75%) in the gastrointestinal tract, although extensive first-pass metabolism cuts its bioavailability to less than 5%. AGM is mostly metabolized by different hepatic cytochromes (CYP450, CYP1A2, and CYP2C9), and the resulting catabolites (mainly 3 hydroxyAGM and 7 desmethyl AGM) are largely eliminated by urinary excretion. Its mean half-life (t1/2) is approximately 2 h [2,7,8,9].

A recent meta-analysis, which included randomized and head-to-head trials of 21 antidepressants for the acute treatment of major depressive disorder (MDD) in adults, highlighted the good efficacy and tolerability of this compound [10]. Moreover, several studies also demonstrated the low incidence of side effects of AGM in anxiety disorders, particularly in GAD [4], and also in obsessive–compulsive disorder (OCD), both in monotherapy and in combination with other drugs [11,12,13,14,15].

Side effects, such as dizziness or nausea, are usually classified as mild or moderate. Of note, the risk of liver injury is dose-dependent and the main risk factors promoting hepatotoxicity include female gender, polypharmacy, and old age [16]. However, there is no agreement on this issue, as different research did not confirm these risk factors [2,17]. Regarding hepatotoxicity, EMA recommends performing liver function tests before starting treatment and then after approximately 3, 6, 12, and 24 weeks. If an elevation of transaminases is detected, these exams should be repeated within 48 h and agomelatine should be discontinued if the increase results in being more than three times above the limit of the normal range [18]. AGM tolerability has been considered throughout clinical trials, as a key benefit for treatment initiation and long-term adherence [1,14]. The lack of early relapse on switching to a placebo supports a minimal discontinuation syndrome [19].

The use of AGM in the pediatric population is limited by the paucity of data. Only one study investigated the efficacy of agomelatine among adolescents with MDD. This randomized, double-blind, multicenter study tested the short-term antidepressant efficacy and safety of AGM (10 mg or 25 mg per day) versus placebo with active control (fluoxetine 10–20 mg depending on symptom severity) after 12 weeks on patients aged 7–17 years. The AGM highest dosage of 25 mg/day resulted in an improvement versus placebo, with a similar effect for fluoxetine in adolescents but not in children. No unexpected safety signals were observed with agomelatine, with no significant weight gain or effect on suicidal behaviors [20].

## 2. Clinical Use of Agomelatine in ADHD and ASD

Here we provide an overview of the recent literature on the use of AGM in patients with ASD and ADHD.

### 2.1. ADHD

There is a limited number of clinical studies on the use of AGM in the ADHD population. We report on six articles. Three of them are single cases and two are clinical trials. All studies included children and adolescents. (Table 1).

In 2012, Niederhofer et al. demonstrated greater efficacy of add-on AGM (25 mg/day) to ongoing treatment with methylphenidate (MPH) or atomoxetine compared to placebo in a sample of 10 boys with ADHD (age range 17–19 years, M:F = 8:2) [7,8]. Patients treated showed less hyperactivity, greater tolerance to frustration, less irritability, and overall better affective modulation. Few side effects were observed, suggesting that AGM is a valid therapeutic option, especially in ADHD patients with sleep disorders (70% ADHD), anxiety, or oppositional–provocative comorbid disorder [21].

The addition of AGM (25 mg/day) to MPH in a 13-year-old female with severe ADHD, insomnia, and dysphoria resulted in clinical improvement and a global enhancement of MPH’s cognitive and behavioral effects. Good tolerability and no drug interactions were reported [22]. More recently (2020), the same authors reported an interesting case of a 15 year old girl with drug-resistant ADHD, who responded only to agomelatine (25 mg/die) [24].

In a cohort of 54 children with ADHD aged between 6 and 15 years of age, a pharmacological study compared the efficacy of MPH (20–30 mg/die depending on weight) with AGM (15–25 mg/day depending on weight) over about 6 weeks. No statistically significant difference emerged between the two randomized groups in terms of efficacy according to the parent and teacher ADHD Rating Scale-IV. Agomelatine was well-tolerated and also improved insomnia [23].

In ADHD patients have reduced neural tissue volume, particularly in the right frontal and parietal cortices, in association with hypo-functioning catecholaminergic circuits in the prefrontal cortex, which may explain the finding of AGM being comparatively effective as MPH for ADHD treatment [25,26].

### 2.2. ASD

The impact of AGM on ASD has been studied in the adult population or in animal models, with the exception of a single case report of a 10 year old child. (Table 2).

In 2015, Nagury emphasized the therapeutic efficacy of AGM on a 10-year-old autistic patient with severe behavioral disorder and insomnia, poorly responsive to aripiprazole and risperidone. Clinical improvement in terms of irritability, hyperactivity, and stereotypes was reported after the administration of AGM (25 mg/day) as monotherapy [28].

In the same year, Ballester conducted a randomized, cross-double-blind, multicenter study of 25 patients with ASD (M:F = 20:5) with an average age of approximately 32 years.

Participants were randomized to receive AGM (25 mg/die) or placebo for 3 months, at two different moments. Functional circadian rhythm markers (i.e., wrist temperature, actimetry, and position [TAP]) and salivary cortisol were measured during a week at the beginning and at the end of each period. Significant differences in improving circadian rhythms in the AGM group were noticed. AGM was effective in improving sleep patterns in ASD compared with placebo [29].

Later, the same group published a cross-sectional, randomized, triple-blind, placebo-controlled study conducted on 23 adult patients with ASD and intellectual disability with associated sleep disturbance. The results show a significant improvement in the sleep structure in the group treated with AGM, in terms of increased total night-time sleep time, sleep stability, and phase correction. The adverse effects were minimal [30].

In animal models of ASD induced by prenatal administration of valproic acid (VPA) in mice (2015), it was observed that the administration of AGM was associated with a reduction in autistic-like behaviors, a decrease in oxidative and nitrosative markers, and inflammation [31].

Previously, in 2011, Niederhofer demonstrated that AGM (25 mg/die) was not more efficient compared to duloxetine (40 mg/day), respectively, in one and two adult patients with ASD and intellectual disability. In addition, no major adverse effects were reported [27].

## 3. Potential Effects of AGM in Neurodevelopmental Disorders

Despite belonging to different and well-defined nosographic entities, neurodevelopmental disorders and, more broadly, psychiatric disorders seem to share common etiopathogenetic mechanisms, with consequent symptomatic overlap [32,33,34,35].

Numerous pieces of evidence show how the pathophysiology of ASD, ADHD, anxiety, and depression is multifactorial, involving different mechanisms such as neuroinflammation, oxidative stress, and glutamatergic dysfunction [33,34,35,36,37,38,39,40,41,42]. These pathways are complementary and strictly interconnected, able to activate a self-amplifying vicious circle [38,39,40,41].

There is currently no specific cure for the symptomatic core of autism, while psychotropic medication aims to alleviate psychiatric and behavioral problems such as aggression, self-injury, impulsivity, hyperactivity, irritability, anxiety, and mood disorders. Benefits have been reported with (i) atypical antipsychotics for aggression, self-injurious behavior, or temper tantrums; (ii) selective serotonin reuptake inhibitors (SSRI) for anxiety and repetitive behaviors; and (iii) psychostimulants or opioid antagonists for hyperactivity [43].

Psychostimulants are still considered the most effective therapy for children, adolescents, and adults with ADHD. The first choice drug is MPH [44,45,46,47]. Among noradrenergic modulators approved for the treatment of ADHD, we find tricyclic antidepressants with secondary amine structures such as desipramine and nortriptyline, alpha-2 adrenergic agonists including clonidine and guanfacine, indirect agonists such as bupropion, and atomoxetine, which is a selective blocker of the reuptake of noradrenaline [45,48].

In general, AGM treatment would aim not only to reduce depressive and anxiolytic symptoms, but also to prevent relapses, chronicity, and complications, and improve social and functional adaptation.

Anxiety and depression co-occur frequently with ADHD and ASD, increasing their clinical severity [49,50,51,52,53,54].

Although available antidepressants significantly ameliorate these disorders, they take several weeks to exert their full efficacy. Furthermore, many patients respond inadequately and co-morbid symptoms are often not well-controlled, leading to problems of poor tolerance, including gastrointestinal disturbances, weight gain, sleep disturbances, sexual dysfunction, and discontinuation effects [55].

### 3.1. Melatoninergic Action

Sleep is a fundamental operating state of the central nervous system and occupies up to a third of the human life span. Human research demonstrated a central role of sleep in mental health, influencing a wide range of cognitive and emotional functions such as memory consolidation, problem-solving, creativity, affective reactivity, and management of interpersonal conflicts [56,57,58,59,60].

Genes associated with circadian rhythm regulation have been found to be related to many mental diseases, and the function of dopaminergic and serotonergic networks has been shown to interplay with circadian and sleep biological mechanisms [61].

Neurobiological balance between arousal and de-arousal is disturbed in most mental disorders and likely represents a basic dimension for brain function [56,62]. From this point of view, it can be assumed that sleep impairment plays a relevant role in facilitating and maintaining mental disorders and that transdiagnostic treatment of sleep disturbance may improve intervention outcomes [61,62].

An intrinsic melatonin deficiency has been supposed to dysregulate sleep architecture [63].

Some studies found abnormal concentrations of melatonin in the urine and blood of patients with neurodevelopment disorders, corresponding to altered circadian release patterns [34,64,65].

Melatonin is an indole hormone that is enzymatically synthesized in the pineal gland from the amino acid tryptophan by N-acetylation and subsequent O-methylation of 5-HT [66]. Its secretion is inhibited by light and regulated by the circadian clock located in the hypothalamic suprachiasmatic nuclei. In humans, this neurohormone is mainly produced in the pineal gland, gastrointestinal tract, and retina, but only melatonin secretion by the pineal gland and retina follows a typical circadian rhythm [67]. At the onset of darkness, reduced retinal input leads to the disinhibition of enzymes responsible for melatonin synthesis [68]. (Figure 1). This increased synthesis leads to the highest nocturnal plasma concentrations of approximately 80 to 120 pg/mL between 2 and 4 h. Then, melatonin levels decrease until daylight onset, with low (10–20 pg/mL) concentrations during the daytime [67]. A study by Sadeh et al. demonstrated that infants with an immature pattern of melatonin secretion showed a delayed peak in melatonin levels, with more fragmented sleep during the night [69].

Due to the low bioavailability of melatonin as a supplement, AGM would represent a better pharmacological strategy for restoring melatoninergic homeostasis and physiological circadian rhythmicity [41,70].

### 3.2. Anti-Inflammatory and Oxidative Stress Action

Melatonin regulates several biological functions since it shows anti-inflammatory, antioxidant, and free radical scavenging properties.

Several authors argue that the depletion of endogenous melatonin could be explained by hyperactivation of the Kynurenines pathway (KP) from tryptophan, with the subsequent reduction in its bioavailability for the synthesis first of serotonin and then of melatonin [71,72,73]. Kynurenines represent a heterogeneous group of neuroactive catabolites, which mediate different pathways such as neuroinflammation, redox homeostasis, and glutamatergic toxicity by acting directly and indirectly on neurotransmitter systems involved in the pathogenesis of psychiatric disorders [72]. (Figure 2).

The shift of tryptophan towards the KP seems to be promoted by high concentrations of pro-inflammatory cytokines (IL-6, INF-y), ROS/NOS, and cortisol, which are associated with neuroinflammation, oxidative stress, and HPA-overactivation [39,41,71,72,73].

Interestingly, polymorphisms localized in genes involved in the tryptophan catabolic pathway may modulate the response to antidepressant treatment. Polymorphism of tryptophan hydroxylase-1 and -2 (TPH1- TPH2) and kynurenine aminotransferase I (KATI) genes may be associated with a lack of response to conventional antidepressant therapy since they can change messenger RNA (mRNA) and protein expression levels or methylation status of promoter regions [74,75,76,77,78].

Studies on animal models highlighted how the administration of AGM may modulate the transduction of mRNA and protein, as well as the expression of genes involved in the tryptophan pathway in the blood and brain structures [74,75]. This potentially may protect the brain from the neurotoxic consequences, due to the conversion of kynurenic acid (KYNA) to quinolinic acid (QUIN).

Furthermore, AGM treatment ameliorated inflammatory responses by decreasing the protein levels of inflammasome components, and by inhibiting microglial activation through the toll-like receptor 4/NOD-like receptor protein 3 (TLR4/NLRP3) signaling pathway [79].

Oxidative stress may be associated with the development of depression, ASD, and ADHD. A recent pre-clinical study shows that chronic administration of AGM modifies the expression level and methylation status of the promoter region of genes involved in oxidative and nitrosative stress. Stressed rats treated with AGM displayed a significantly lower glutathione peroxidase 4 (Gpx4) level in the hypothalamus [74]. Additionally, for the evaluation of the effect of AGM on oxidative stress and inflammation, glutathione (GSH), malondialdehyde (MDA), tumor necrosis factor (TNF), and interleukin-6 (IL-6) levels were analyzed in immortalized mouse hippocampal neuronal cell HT-22 and in hippocampal tissues in male rats. AGM significantly attenuated oxidative stress and inflammation due to the cisplatin insult in vitro and in vivo, and ameliorated the neuronal pathology in the hippocampus, which is strongly related to cognition and memory [80].

### 3.3. Neurotrophic, Anti-Glutamatergic, and Anxiolytic Actions

One of the most relevant effects of melatonin and, therefore, of AGM, is the activation of the gene expression of the neurotrophic factor brain-derived neurotrophic factor (BDNF) in the prefrontal cortex, hippocampus, subventricular area, and olfactory tubercle [42,81]. BDNF is responsible for neurogenesis and neuronal trophism, underpinning neuronal plasticity [81]. Neurogenesis (especially hippocampal neurogenesis) has been implicated in cognitive processes such as learning, memory, pattern separation, and cognitive flexibility [82,83]. Abnormal neuronal plasticity leads to an inability to adapt to stressful stimuli, with reduced resilience and dysfunctional behaviors [81,84,85,86]. Furthermore, AGM would increase the expression of the activity-regulated cytoskeleton-associated protein (ARC) in the prefrontal cortex [87,88]. ARC regulates glutamatergic synapse plasticity and its expression is induced in a number of brain regions following emotionally relevant experiences, including exposure to novelty, environmental enrichment, and stressful experiences. It is also involved in other cognitive functions such as memory consolidation, spatial learning, and memory, and fear learning [87].

AGM has been shown to modulate glutamatergic neurotransmission in regions associated with mood and cognition. More precisely, AGM reduces the stress-induced release of glutamate in the prefrontal and frontal cortex [38]. In predisposed subjects, exposure to intense and chronic stress excessively activates excessively the hypothalamic–pituitary–adrenal (HPA) axis, causing a high release of ACTH and cortisol [88,89]. High concentrations of glucocorticoids lead to an excessive release of glutamate, with the imbalance between inhibition/excitation of the neuronal circuits and consequent glutamatergic excitotoxicity [89].

Excitotoxicity promotes abnormal neurogenesis and neuronal plasticity, as well as dysfunction in the default mode network, and visual and auditory systems, with the onset of cognitive deficits, general dyspraxia, behavioral changes, social dysfunction, and EEG abnormalities/seizures [63,90].

In animal studies, chronic AGM administration (3 weeks) increased cell proliferation and neurogenesis in the ventral dentate gyrus (a region implicated in fear-related behaviors) and prevented transcription of the interleukin-1β (IL-1β) and metabotropic glutamatergic receptor (mGluR) genes [91].

AGM also displayed robust anxiolytic properties in rats [1]. Early life adversity (e.g., social isolation rearing) has profound effects on neurodevelopment, altering 5-HT and other neurotransmitters, leading to anxiety, depression, and antisocial behavior in later stages [92]. Social isolation rearing is known to destabilize circadian rhythms while altered circadian fluctuation of different neuropeptides, such as vasopressin, oxytocin, and corticosterone has been described in chronic anxiety states [93]. In rats, social isolation rearing is associated with reduced plasma corticosterone and oxytocin and increased vasopressin.

Given the chronobiotic effects of AGM through simultaneous stimulation of MT1/MT2 and blockade of 5-HT2C receptors in the suprachiasmatic nucleus, during chronic stress, AGM may regulate oxytocin and vasopressin release by resynchronizing related neuropeptide rhythm disorders [94]. Thus, the anxiolytic effect of AGM could also be attributed to a decrease in vasopressin and an increase in oxytocin levels [92,93].

### 3.4. 5-HT2C Antagonist Action

Dopaminergic and noradrenergic pathway dysfunction in the frontal cortex represents one of the neurobiological mechanisms underlying psychiatric disorders. The frontal cortex plays a fundamental role in cognitive processes and the regulation of behavior and emotions [94,95,96].

Impairment in this area leads to loss of motivation and disinhibition, considered clinical key features of depression, ADHD, ASD, and some anxiety disorders [96,97,98].

Neuroimaging techniques prove that symptomatologic overlap between those disorders is linked to dysfunction in specific neuroanatomical areas. In depressed patients, the basal activity of the dorsolateral prefrontal cortex is reduced; in the ADHD population, the frontal cortex and some basal ganglia (caudate nucleus and the pale globe) are smaller [99,100,101], and characterized by a slower activation with less oxygen consumption [102]. In ASD patients, synaptic and neurotransmitter dissociation between the prefrontal cortex and other brain regions has been noted, which, in turn, has been correlated to social and communication impairment [103,104,105,106].

AGM inhibits the release of dopamine and norepinephrine, by blocking 5-HT2C presynaptic receptors, enhancing its transmission in the prefrontal cortex [48]. The increase in norepinephrine and dopamine availability in the frontal cortex acts both as antidepressant and psychostimulant.

This innovative mechanism for an antidepressant provides more physiological activation of the areas of executive functions such as working memory and decision-making processes, improving, theoretically, ASD and ADHD functioning. In particular, AGM seems to ameliorate intentional cognition and discriminatory attention, leading to less irritability and better affective regulation, as shown in patients on treatment [107,108].

## 4. Potential Therapeutic Role of AGM against SARS-CoV-2

Melatonin has been recently identified among the top five molecules with potential anti-SARS-CoV-19 action [109].

It has been hypothesized that melatonin significantly inhibits inflammasome stimulation, which could indirectly reduce the intensity of the cytokine storm following infection [110]. Moreover, it can promote the restoration of circadian rhythm and mitochondrial metabolism and can regulate the cellular oxidative status, favoring cell survival in the lung under stress and inflammatory conditions [111,112].

In a one-silico study, it was demonstrated that melatonin could act as a SARS-CoV-2 main protease (Mpro) inhibitor [113]. Another preclinical study showed that melatonin, AGM, and ramelteon can bind to host cell angiotensin-converting enzyme 2 receptors (ACE 2) and viral receptor-binding domain (RBD) [112,114,115], preventing viral entry into the host cells [112]. More recently, in transgenic mice expressing human ACE 2 receptor (K18-hACE 2), strongly susceptible to SARS-CoV-2 infection, daily melatonin, AGM, or ramelteon administration delayed the occurrence of severe clinical outcomes, with the improvement in survival, especially with a high melatonin dose [116].

By restoring antioxidant and inflammatory status and sleep patterns in COVID-19 patients, AGM could serve as an adjuvant in COVID-19 disease management [117]. It is to be reiterated that it could also be used as an antidepressant to manage the psychiatric complications of COVID-19 infection [112].

## 5. Conclusions

Comorbidity is the rule and not the exception in mental illnesses, making it important to spread the focus from specific disorders to psychobiological mechanisms that cut across mental disorders.

In the pediatric population, the strict categorical approach to psychopathology is much more difficult than in the adult population. During childhood and adolescence, psychiatric symptoms appear as much more nuanced and variable, in line with the different ages of development and the immaturity of the internal psychic systems.

Therefore, the idea of a symptomatologic continuum among mental disorders constitutes an increasingly pressing concept. According to what has been said, the use of multi-target drugs that act on specific symptoms, modulating simultaneously the pathological background, would allow not only the customization of therapy according to the patient’s need, but also the treatment of different pathological conditions. With its synergistic activity as a melatoninergic agonist and 5-HT2C antagonist, AGM acts as an antidepressant, a psychostimulant, and a promoter of neuronal plasticity, regulating cognitive, affective symptoms, and resynchronizing circadian rhythms in patients with autism, ADHD, anxiety, and depression. It is considered to have a favorable side-effect profile, without the weight gain, sexual side-effects, or discontinuation syndrome seen with traditional antidepressants such as SSRI/SNRIs.

Given its good tolerability and good compliance, AGM can be potentially administered to teenagers and children. To bypass the current gap of data on the use of AGM in developmental patients, controlled clinical trials are needed. The high frequency with which anxiety, depression, and sleep disorders co-occur in ASD and ADHD, in conjunction with their unique impact on functioning, adult outcome, and quality of life, is critical to consider how to best prevent and treat these disorders among children and adolescents. Also, considering the historical moment and the personal and family upheaval that the COVID-19 emergency has created and is still creating, it is not surprising to detect an exponential growth of affective, cognitive, and behavioral symptoms, as well as sleep disturbances, among children and adolescents with neurodevelopmental disorders. This phenomenon could be presumably correlated to pandemic-related stress and higher anxiety levels in vulnerable adolescents and young adults [118,119]. In this scenario, using a versatile and tolerable drug such as AGM would represent an innovative opportunity to improve the therapeutic perspective of chronic neurodevelopmental disorders. Modulation of neurocircuits in developmental age would improve clinical outcomes, preventing chronicity of therapy and, of course, bringing a reduction in dysmetabolic side effects.

## Figures and Tables

**Figure 1 brainsci-13-00734-f001:**
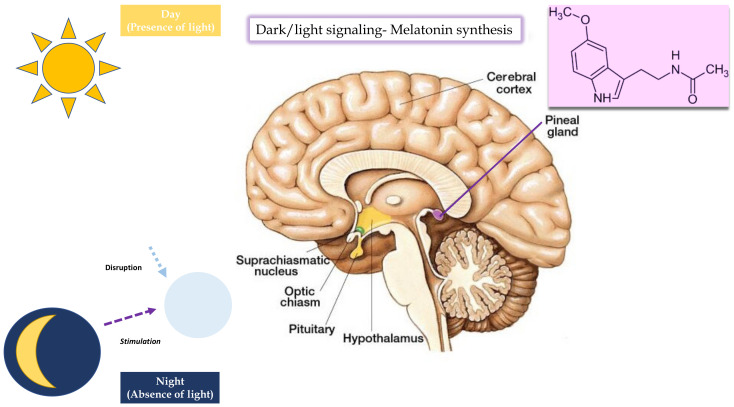
Dark/light signaling, melatonin synthesis.

**Figure 2 brainsci-13-00734-f002:**
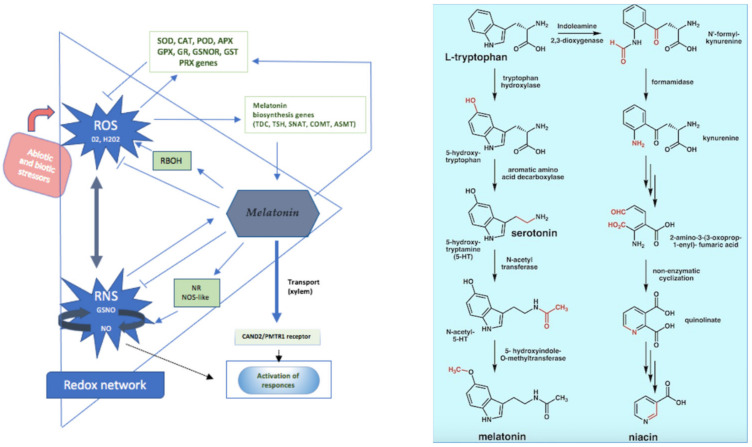
Melatonin antioxidant properties and tryptophan metabolic pathway.

**Table 1 brainsci-13-00734-t001:** Clinical use of AGM in ADHD patients.

Authors	Year	Article Type	Age (Year)	Number of Patients	Gender	Diagnosis/Comorbidity	Dose of Agomelatine
Niederhofer H., et al. [21]	2012	Placebo-controlled study	17–19 years old	10	M:F = 8:2	Severe ADHD	25 mg/day
Naguy A. and Al-tajali A. [22]	2015	Case report	13 years old	1	F	Severe ADHD	25 mg/day
Salardini E., et al. [23]	2016	Double-blind randomized controlled trial	6–15 years old	54	Not available	Severe ADHD	15 mg/day in patients with weight ≥30 kg and 25 mg/day in patients with weight ≥45 kg
Naguy A. and Alamiri B. [24]	2020	Case report	15 years old	1	F	Severe ADHD/migraine	25 mg/day

ADHD = attention deficit hyperactivity disorder; ID = intellectual disability.

**Table 2 brainsci-13-00734-t002:** Clinical use of AGM in ASD patients.

Authors	Year	Article Type	Age (Year)	Number of Patients	Gender	Diagnosis/Comorbidity	Dose of Agomelatine
Niederhofer H., et al. [27]	2011	Case report: 10 week clinical trial	Adults	2:1	M	Severe ASD/ID	25 mg/day
Naguy A. and Ali Al Tajali [28]	2015	Case report	10 years old	1	M	Severe ASD/behavioral disorder and insomnia	25 mg/day
Ballester P., et al. [29]	2015	Randomized, cross-double-blind, multicenter study	30–32 years old	25	M:F = 20:5	Severe ASD	25 mg/day
Ballester P., et al. [30]	2019	Cross-sectional, randomized, triple-blind, placebo-controlled study	35 ± 12 years old	23	M:F = 19:4	ASD/ID and sleep disturbance	25 mg/day
Kumar H., et al. [31]	2015	Study on animal models	/	/	/	Autism VPA-induced	/

ASD = autism spectrum disorder; ID = intellectual disability; VPA = valproic acid.

## Data Availability

Not applicable.

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
