# Peer review of "Agomelatine: A Potential Multitarget Compound for Neurodevelopmental Disorders"

_brainsci, 2023, doi:10.3390/brainsci13050734_

Round 1

Reviewer 1 Report

Here, Savino et al. reviewed and summarized previous works on AGM as an atypical antidepressant for treating neurodevelopment diseases. This review provides decent summary and comments on the published works. However, there are a few minor issues needs to be added or changed: 

1. The acronym "AGM" needs to be explained in full name "Agomelatine" at the place it showed up for the first time; 

2. Typo in the line 30: "AGM" should be added before "is".

Author Response

Dear Reviewer,

thank you for your positive opinions and relevant comments.

As you suggested, we used “Agomelatine” in full name in the title, in the abstract, and manuscript when it shows up for the first time. Furthermore, we add  AGM  at line 30  in the abstract.

  • Title: “Agomelatine: a potential multitarget compound for neurodevelopmental disorders
  • Abstract: Agomelatine (AGM) is one of the latest atypical antidepressants, indicated exclusively for the treatment of depression in adults. AGM belongs to the pharmaceutical class of "MASS" (Melatoninergic Agonist and Selective Serotonin Antagonist) as it acts both as a selective agonist of melatonin receptors MT1 and MT2, and as a selective antagonist of 5-HT2C/5-HT2B receptors. AGM is involved in the resynchronization of interrupted circadian rhythms with beneficial effects on sleep patterns, while antagonism on serotonin receptors increases the availability of norepinephrine and dopamine in the prefrontal cortex, with an antidepressant and nootropic effect. …

  • Introduction: Agomelatine (AGM) is a melatonin analog with antidepressant properties indicated for the treatment of depression in adults…..

Reviewer 2 Report

This is a well-structured review article. The main question addressed by this review is the potential use of the recent atypical antidepressant AGM in neurodevelopmental disorders.

The introduction gives the background of this study as it briefly describes AGM, its pharmacological characteristics and the current literature data regarding its use in ASD and ADHD.

Then the authors describe AGM actions as a melatoninergic, anti-inflammatory, neurotrophic, antiglutamatergic, 5-HT2C antagonist and potentially anti-SARS-COV2 factor.

The “conclusions” section is well written, summarizing and discussing the main findings of the review.

References are relative to the subject and adequate in number.

English language and style are generally fine but there are some minor issues that need to be addressed. For example, in line 49 the word “antidepressants” should be corrected to “antidepressant” and in line 66 a “,” should be added after the term “In addition”.

Author Response

Dear reviewer,

We would like to thank you for your positive opinions and comments.

 As you kindly suggested we have fixed mistakes you found (in line 49 we changed “antidepressants” with “antidepressant”, and we added a comma at line 66 then in addition).

Moreover, we revised English language as you will find in manuscript in red.

Thank you again for your revision.